# A Generative Model for Text Control in Minecraft

**Shalev Lifshitz** [* 1 2] **Keiran Paster** [* 1 2] **Harris Chan** [† 1 2] **Jimmy Ba** [1 2] **Sheila McIlraith** [1 2]

## Abstract

Constructing AI models that respond to text instructions is challenging, especially for sequential decision-making tasks. This work introduces an instruction-tuned Video Pretraining (VPT) model for Minecraft™ called STEVE-1, demonstrating that the unCLIP approach, utilized in DALL•E 2, is also effective for creating instruction-following sequential decision-making agents. STEVE-1 is trained in two steps: adapting the pretrained VPT model to follow commands in MineCLIP's latent space, then training a prior to predict latent codes from text. This allows us to finetune VPT through self-supervised behavioral cloning and hindsight relabeling, bypassing the need for costly human text annotations. By leveraging pretrained models like VPT and MineCLIP and employing best practices from text-conditioned image generation, STEVE-1 **costs just $60 to train and can follow a wide range of short-horizon open-ended text and visual instructions in Minecraft**. STEVE-1 sets a new bar for open-ended instruction following in Minecraft with low-level controls (mouse and keyboard) and raw pixel inputs, far outperforming previous baselines. We provide experimental evidence highlighting key factors for downstream performance, including pretraining, classifier-free guidance, and data scaling. All resources, including our model weights, training scripts, and evaluation tools are made available for further research.

## 1. Introduction

The ability to use text instructions to control and interact with powerful AI models has made these models accessible and customizable for the masses. Such models include

---

[*]Equal contribution , [†]Core contribution, [1]Department of Computer Science, University of Toronto, Toronto, Canada [2]Vector Institute for Artificial Intelligence, Toronto, Canada. Correspondence to: Shalev Lifshitz <shalev.lifshitz@mail.utoronto.ca>, Keiran Paster <keirp@cs.toronto.edu>.

Interactive Learning with Implicit Human Feedback Workshop at ICML 2023.

ChatGPT (OpenAI, 2022), which can respond to messages written in natural language and perform a wide array of tasks, and Stable Diffusion (Rombach et al., 2022), which turns natural language into an image. While those models cost anywhere from hundreds of thousands to hundreds of millions of dollars to train, there has been an equally exciting trend whereby powerful open-source foundation models like LLaMA (Touvron et al., 2023) can be finetuned with surprisingly little compute and data to become instruction-following (e.g., (Taori et al., 2023; Chiang et al., 2023)).

In this paper, we study whether such an approach could be applicable to sequential decision-making domains. Unlike in text and image domains, diverse data for sequential decision-making is very expensive and often does not come with a convenient "instruction" label like captions for images. We propose to instruction-tune pretrained generative models of behavior, mirroring the advancements seen in recent instruction-tuned LLMs like Alpaca (Taori et al., 2023).

In the past year, two foundation models for the popular open-ended video game Minecraft™ were released: a foundation model for behavior called VPT (Baker et al., 2022) and a model aligning text and video clips called MineCLIP (Fan et al., 2022). This has opened up an intriguing avenue to explore fine-tuning for instruction-following in the sequential decision-making domain of Minecraft. VPT was trained on 70k hours of Minecraft gameplay, so the agent already has vast knowledge about the Minecraft environment. However, just as the massive potential of LLMs was unlocked by aligning them to follow instructions, it is likely that the VPT model has the potential for general, controllable behavior if it is finetuned to follow instructions. In particular, our paper demonstrates a method for fine-tuning VPT to follow short-horizon text instructions with only $60 of compute and around 2,000 instruction-labeled trajectory segments.

Our method draws inspiration from unCLIP (Ramesh et al., 2022), the approach used to create the popular text-to-image model DALL•E 2. In particular, we decompose the problem of creating an instruction-following Minecraft agent into two models: a VPT model finetuned to achieve visual goals embedded in the MineCLIP latent space, and a prior model that translates text instructions into MineCLIP visual embeddings. We finetune VPT using behavioral cloning with self-supervised data generated with hindsight relabeling (Andrychowicz et al., 2017), avoiding the use of expensive

text-instruction labels in favor of visual MineCLIP embeddings. We apply unCLIP with classifier-free guidance (Ho and Salimans, 2022) to create our agent called STEVE-1, which sets a new bar for open-ended instruction following in Minecraft with low-level controls (mouse and keyboard) and raw pixel inputs, far outperforming the baseline set by Baker et al. (2022).

Our main contributions are as follows:

- We create STEVE-1, a Minecraft agent that can follow open-ended text and visual instructions with a high degree of accuracy. We perform extensive evaluations of our agent, showing that it can perform a wide range of short-horizon tasks[1] in Minecraft. For longer-horizon tasks like crafting and building, we show that a basic version of prompt chaining can dramatically improve performance.

- We describe our method for creating STEVE-1 using only $60 of compute, showing that unCLIP (Ramesh et al., 2022) and classifier-free guidance translate well to the sequential decision-making domain and are essential for strong performance.

- We release model weights for STEVE-1 as well as training scripts and evaluation code in order to foster more research into instructable, open-ended sequential decision-making agents.[2]

## 2. Related Work

**Minecraft as a Test-bed for AI**   Minecraft has gained popularity as a benchmark for AI research due to its complex and dynamic environment, making it a rich test-bed for reinforcement learning and other AI methods (e.g., (Johnson et al., 2016; Guss et al., 2019; Fan et al., 2022; Hafner et al., 2023; Nottingham et al., 2023; Wang et al., 2023; Malato et al., 2022; Cai et al., 2023)). We leverage the MineRL environment (Guss et al., 2019) to research the creation of agents that can follow open-ended instructions in complex visual environments using only low-level actions (mouse and keyboard). We build STEVE-1 on top of two recent foundation models. In order to align text and videos, we use MineCLIP (Fan et al., 2022), a CLIP (Radford et al., 2021) model trained on paired web videos of Minecraft gameplay and associated captions. To train STEVE-1's policy, we fine-tune VPT (Baker et al., 2022), a foundation model of Minecraft behavior that is pretrained on 70k hours

of web videos of Minecraft along with estimated mouse and keyboard actions. Several prior works (Volum et al., 2022; Wang et al., 2023) have explored the use of LLMs in creating instructable Minecraft agents. These works typically use LLMs to make high-level plans that are then executed by lower-level RL (Nottingham et al., 2023; Wang et al., 2023) or scripted (PrismarineJS and Others, 2023) policies. Since STEVE-1 is a far more flexible low-level policy, the combination of STEVE-1 with LLMs is a promising direction for future work. Fan et al. (2022) introduced an agent trained using RL on 12 different tasks and conditioned on MineCLIP-embedded text-prompts. However, this agent failed to generalize beyond the original set of tasks.

**Foundation Models for Sequential Decision-Making**
Foundation models which are pretrained on vast amounts of data and then finetuned for specific tasks have recently shown great promise in a variety of domains including language (Brown et al., 2020; Chowdhery et al., 2022; Touvron et al., 2023), vision (Ramesh et al., 2022; Caron et al., 2021; Radford et al., 2021), and robotics (Brohan et al., 2022; Shridhar et al., 2022; Jiang et al., 2022; Nair et al., 2022; Xiao et al., 2022). GATO (Reed et al., 2022) and RT-1 (Brohan et al., 2022) have demonstrated the potential of training transformers to perform both simulated and real-world robotic tasks. With the exception of Kumar et al. (2023), which uses Q-learning, the vast majority of cases (Lee et al., 2022; Brohan et al., 2022; Reed et al., 2022) where deep learning has been scaled to large, multitask offline-RL datasets have used supervised RL. Supervised RL (e.g., (Paster et al., 2020; Ghosh et al., 2021; Chen et al., 2021)) works by framing the sequential decision-making problem as a prediction problem, where the model is trained to predict the next action conditioned on some future outcome. While these approaches are simple and scale well with large amounts of compute and data, more work is needed to understand the trade-offs between supervised RL and Q-learning or policy gradient-based methods (Paster et al., 2022a;b; Brandfonbrener et al., 2022; Strupl et al., 2022). Recent works explore the use of hindsight relabeling (Andrychowicz et al., 2017) using vision-language models (Radford et al., 2021; Alayrac et al., 2022) to produce natural language relabeling instructions. DIAL (Xiao et al., 2022) finetunes CLIP (Radford et al., 2021) on human-labeled trajectories, which is then used to select a hindsight instruction from a candidate set. Sumers et al. (2023) uses Flamingo (Alayrac et al., 2022) zero-shot for hindsight relabeling by framing it as a visual-question answering (VQA) task. In contrast, STEVE-1 relabels goals using future trajectory segment embeddings given by the MineCLIP (Fan et al., 2022) visual embedding.

**Text-Conditioned Generative Models**   There has been a recent explosion of interest in text-to-X models, includ-

---

[1]Short-horizon tasks require few steps: e.g., go to a tree and chop it down, dig a hole. Long-horizon tasks take many steps: e.g., craft complex recipes from scratch, build a house.

[2]Model weights, training code, videos, and an interactive demo script are hosted on our project webpage at https://sites.google.com/view/steve-1.

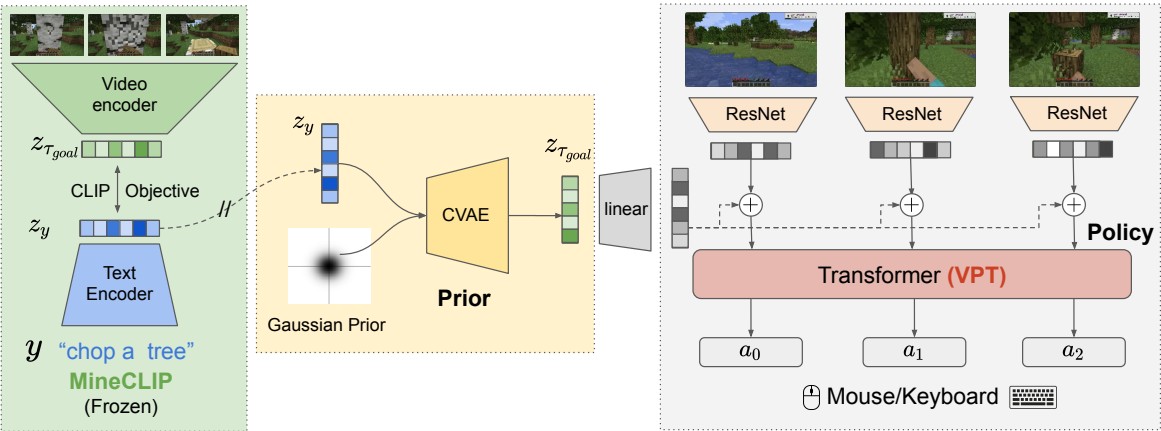

Figure 1. Like unCLIP (Ramesh et al., 2022), our approach involves two models. First, we train the *policy* by finetuning VPT to achieve goals given by pretrained MineCLIP (Fan et al., 2022) visual embeddings using our gameplay dataset. Second, for the *prior* model, we train a CVAE (Sohn et al., 2015) to sample MineCLIP visual embeddings given a text prompt. The combination of these two models enables our agent to follow text and visual instructions.

ing text-to-image (e.g., (Ramesh et al., 2022; Saharia et al., 2022; Rombach et al., 2022)), text-to-3D (e.g., (Jun and Nichol, 2023; Lin et al., 2023)), and even text-to-music (e.g., (Agostinelli et al., 2023)). These models are typically either autoregressive transformers modeling sequences of discrete tokens (Vaswani et al., 2017; Brown et al., 2020) or diffusion models (Ho et al., 2020). Most related to our work is unCLIP, the method used for DALL•E 2 (Ramesh et al., 2022). unCLIP works by training a generative diffusion model to sample images from CLIP (Radford et al., 2021) embeddings of those images. By combining this model with a prior that translates text to visual CLIP embeddings, unCLIP can produce photorealistic images for arbitrary text prompts. unCLIP and many other diffusion-based approaches utilize a technique called classifier-free guidance (Ho and Salimans, 2022), which lets the model trade-off between mode-coverage and sample fidelity post-training. We utilize the basic procedure of unCLIP and classifier-free guidance for training STEVE-1.

## 3. Method

Inspired by the rapid recent progress in instruction-tuning Large Language Models (LLMs), we choose to leverage the recently released Video Pretraining (VPT) (Baker et al., 2022) model as a starting point for our agent. Since VPT was trained on 70k hours of Minecraft gameplay, the agent already has vast knowledge about the Minecraft environment. However, just as the massive potential of LLMs was unlocked by aligning them to follow instructions, it is likely that the VPT model has the potential for general, controllable behavior if it is finetuned to follow instructions. In this work, we present a method for finetuning VPT to follow natural, open-ended textual and visual instructions,

which opens the door for a wide range of uses for VPT in Minecraft.

Our approach is inspired by unCLIP, the method behind the recent text-to-image generation model, DALL•E 2 (Ramesh et al., 2022). Our goal is to create a generative model of behavior in Minecraft conditioned on text instructions $y$. To do so, we utilize a dataset of Minecraft trajectory segments, some of which contain instruction labels $y$: $[(\tau_1, y_1), (\tau_2, y_2), \ldots, (\tau_n, \emptyset)]$ where $\tau$ is a trajectory of observations and actions. We also employ a pretrained CLIP model called MineCLIP (Fan et al., 2022), which generates aligned latent variables $z_{\tau_{t:t+16}}, z_y$, where $z_{\tau_{t:t+16}}$ is an embedding of any 16 consecutive timesteps from the trajectory. MineCLIP is trained using a contrastive objective on pairs of Minecraft videos and transcripts from the web. For simplicity of notation, we refer to the MineCLIP embedding of the last 16 timesteps of a trajectory segment as $z_{\tau_{goal}}$. Like unCLIP (Ramesh et al., 2022), we utilize a hierarchical model consisting of a prior and a policy:

- A *prior* $p(z_{\tau_{goal}}|y)$ that produces a latent variable $z_{\tau_{goal}}$ conditioned on a text instruction $y$.

- A *policy* $p(\tau|z_{\tau_{goal}})$ that produces a trajectory conditioned on a latent variable $z_{\tau_{goal}}$.

These two models can then be combined to produce a generative model of behaviors conditioned on text instructions:

$$p(\tau|y) = p(\tau, z_{\tau_{goal}}|y) = p(z_{\tau_{goal}}|y)p(\tau|z_{\tau_{goal}}) \quad (3.1)$$

### 3.1. Policy

To learn our policy, we finetune VPT, a foundation model of Minecraft behaviors $p_\theta(\tau)$ trained on 70k hours of Minecraft

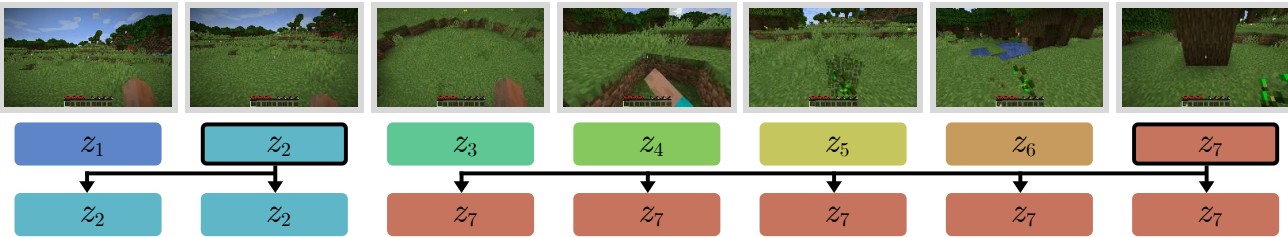

*Figure 2.* To create goal-conditioned data for finetuning, we randomly select timesteps from episodes and use hindsight relabeling to set the intermediate goals for the trajectory segments to those visual MineCLIP embeddings. This self-supervised data teaches the agent which actions lead to which states.

gameplay videos. Specifically, VPT consists of a ResNet (He et al., 2016) that processes frames of dimension $128 \times 128 \times 3$, and a Transformer-XL (Dai et al., 2019) which processes the frame representations and autoregressively predicts the next action using the joint hierarchical action space described in Baker et al. (2022). In order to modify the architecture to condition on goal information, we add an affine transformation of $z_{\tau_{\text{goal}}}$ to the output of the ResNet before passing it to the transformer:

$$
\begin{aligned}
\text{Process Frames:} &\quad \text{ResNet}_\theta(o_t) \rightarrow x_t \\
\text{[+ Goal Embedding]:} &\quad x_t \rightarrow x_t + W_\theta z_{\tau_{\text{goal}}} + b_\theta \\
\text{Predict Actions:} &\quad \text{TransXL}_\theta(x_t, \dots, x_{t+T}) \rightarrow a_{t+T}
\end{aligned}
$$

In order to finetune VPT to condition on goals, we finetune the model using a method inspired by supervised RL approaches like Decision Transformer (Chen et al., 2021), GLAMOR (Paster et al., 2020), and GCSL (Ghosh et al., 2021). We use a modification of hindsight relabeling which we call **packed hindsight relabeling** (see Figure 2) to generate a new dataset of trajectories with goals pulled from future states that periodically switch. Specifically, our method to generate this dataset consists of two steps:

1. Given a trajectory $\tau$ with $T$ timesteps, randomly generate indices to select goals from: $i_1, i_2, \dots, i_n$. These indices are chosen by starting at the first timestep and repeatedly sampling a new timestep by adding a random value to the previous timestep. This ensures that the data reflects that some goals may take longer to achieve than others.
2. For each chosen goal at timestep $i_j$, set the goals for timesteps $i_{j-1} + 1, \dots, i_j$ to be the goal at timestep $i_j$, denoted $z_{\tau_{i_j}}$.

Our final dataset $\mathcal{D}_{\text{relabeled}}$ consists of observation sequences $(o_1, \dots, o_T)$, action sequences $(a_1, \dots, a_T)$, and packed hindsight relabeled goals $(z_1, \dots, z_T)$. We then finetune VPT on this dataset using a supervised loss to predict eacg action autoregressively using a causal attention mask:

$$
\mathcal{L}_{\text{policy}}(\theta) = \mathbb{E}_{\mathcal{D}_{\text{relabeled}}}[-\log p_\theta(a_t|o_{1\dots t}, z_{1\dots t})] \quad (3.2)
$$

## 3.2. Prior

In order to condition not only on embeddings of visual goals but on latent goals, we need the prior, a model that produces a latent variable $z_{\tau_{\text{goal}}}$ conditioned on a text instruction $y$. Our model is a simple conditional variational autoencoder (CVAE) (Sohn et al., 2015; Kingma and Welling, 2014) with a Gaussian prior and a Gaussian posterior. Rather than learn to condition directly on text, we choose to condition on frozen text representations from MineCLIP $z_y$. Both the encoder and decoder of our CVAE are parameterized as two-layer MLPs with 512 hidden units and layer normalization (Ba et al., 2016). We train the model on our dataset, for which we have text labels $\mathcal{D}_{\text{labels}}$ using the following loss:

$$
\begin{aligned}
\mathcal{L}_{\text{prior}}(\phi) =& \mathbb{E}_{(z_{\tau_{\text{goal}}}, z_y) \sim \mathcal{D}_{\text{labels}}} \Big[ \text{KL}(q_\phi(z_{\tau_{\text{goal}}}|z_y) \| p(z_{\tau_{\text{goal}}})) \\
& - \mathbb{E}_{c \sim q_\phi(z_{\tau_{\text{goal}}}|z_y)} \big[ \log p_\phi(z_{\tau_{\text{goal}}}|c, z_y) \big] \Big]
\end{aligned}
$$

## 3.3. Datasets

To train our policy, we gather a gameplay dataset with 54M frames ($\approx$ 1 month at 20FPS) of Minecraft gameplay along with associated actions from two sources: contractor gameplay and VPT-generated gameplay. To train our prior, we use a dataset of text-video pairs gathered by humans and augmented using the OpenAI API `gpt-3.5-turbo` model (OpenAI, 2022) and MineCLIP. See Appendix C for more detailed information about our datasets.

**OpenAI Contractor Dataset** We use 39M frames sourced from the contractor dataset which VPT (Baker et al., 2022) used to train its inverse dynamics model and finetune its policy. The dataset was gathered by hiring human contractors to play Minecraft and complete tasks such as house building or obtaining a diamond pickaxe. During gameplay, keypresses and mouse movements are recorded. We use the same preprocessing as VPT, including filtering out null actions.

**VPT-Generated Dataset** We generate an additional dataset of 15M frames by generating random trajectories us-

ing the various pretrained VPT agents. The diversity of this dataset is improved by randomly switching between models during trajectories (Paster et al., 2022b), randomly resetting the agent's memory, and randomly turning the agent to face a new direction.

**Text-Video Pair Dataset**   To train our prior model, we also manually gather a dataset of 2,000 text instructions paired with 16-frame videos sampled from our gameplay dataset. We augment this dataset by using the alignment between text and video embeddings from MineCLIP. For each text instruction, we find the top $k$ most similar gameplay segments in our dataset and use the corresponding 16-frame video as additional training data. For augmentation, we also add 8,000 text-instructions generated by the OpenAI API `gpt-3.5-turbo` model (OpenAI, 2022), in addition to our 2,000 hand-labeled instructions.

### 3.4. Inference

At inference time, we use the prior to sample a latent goal $z_{\tau_{\text{goal}}}$ from the text instruction $y$. We then use the policy to autoregressively sample actions $a_t$ conditioned on the observation history $o_{1...t}$ and the latent goal $z_{\tau_{\text{goal}}}$. Similar to the observation in Appendix I of Baker et al. (2022), even with conditioning the policy often fails to follow its instruction and simply acts according to its prior behavior. To mitigate this, we borrow another trick used in image generation models: classifier-free guidance. Specifically, during inference we simultaneously compute logits for the policy conditioned on the goal $f(o_t, \ldots, o_{t+1}, z_{\tau_{\text{goal}}})$ and for the unconditional policy $f(o_t, \ldots, o_{t+1})$. We then compute a combination of the two logits using a $\lambda$ parameter to trade-off between the two:

$$\text{logits} = (1+\lambda) \underbrace{f_\theta(o_t, \ldots, o_{t+1}, z_{\tau_{\text{goal}}})}_{\text{conditional logits}} - \lambda \underbrace{f_\theta(o_t, \ldots, o_{t+1})}_{\text{unconditional logits}}$$

By setting a higher value of $\lambda$, we can encourage the policy to follow actions that are more likely when conditioned on the goal and, as demonstrated in Section 4.5, this significantly improves performance. Also, in order to train the policy to generate these unconditional logits, we occasionally dropout the goal embedding $z_{\tau_{\text{goal}}}$ from the policy's input (with probability 0.1). This lets us generate both the conditional and unconditional logits using the same model with batch processing at inference time.

### 3.5. Evaluation

Evaluating the performance of our agent is a challenging task due to the wide variety of instructions that are possible and the difficulty of evaluating whether the agent has successfully achieved its task. We compute *programmatic metrics* by monitoring the agent's travel distance and early-

game item collection (log, seed, and dirt). We also compute automatic *MineCLIP metrics* to evaluate the agent's capability level by recording the minimum cosine distance between the (text or visual) goal embedding and the MineCLIP visual embedding at any timestep during an episode. See Appendix D for more details about our evaluation metrics.

## 4. Results

In our experiments, we aim to answer the following:

1. How well does STEVE-1 perform at achieving both text and visual goals in Minecraft?

2. How does our method scale with more data?

3. What choices are important for the performance of our method?

### 4.1. Training Setup

We base our implementation off of the official VPT code-base[3]. The main STEVE-1 is trained using Pytorch (Paszke et al., 2019) distributed data parallel on four A40 GPUs for 160M frames, or just under three epochs of our gameplay dataset. Hyperparameters are selected to match those in Baker et al. (2022) with the exception of learning rate, which we set to 4e-5. Our models are optimized using AdamW (Loshchilov and Hutter, 2019). See Table 3 for a full list of hyperparameters.

### 4.2. Performance on Textual and Visual Goals

Due to the hierarchical nature of our model, we can evaluate the performance of our agent at achieving either text or visual goals simply by choosing whether to use the prior to condition on text or bypass the prior and condition on a MineCLIP video embedding directly. We first tested our model on a set of 11 tasks that are achievable within the first 2.5 minutes of gameplay and which do not require multiple steps to complete (e.g., chop a tree or dig a hole, but not build a house). A complete list of the tasks and prompts we used for evaluation can be found in Table 4 in the appendix. To select visual goals for testing each of the evaluation tasks, we implemented a tool that searches through 10% of our gameplay dataset by finding the closest 16-frame videos to a given text prompt. We then manually selected a 16-frame video that clearly demonstrates the task being completed and use the corresponding MineCLIP video embedding as the goal embedding for that task. Screenshots of these visual goals can be found in Figure 14 in the appendix.

In Figure 3, we compare the performance of our text and visual-conditioned agents with the unconditional VPT agent across our programmatic tasks. We find that when given

---

[3]https://github.com/openai/Video-Pre-Training

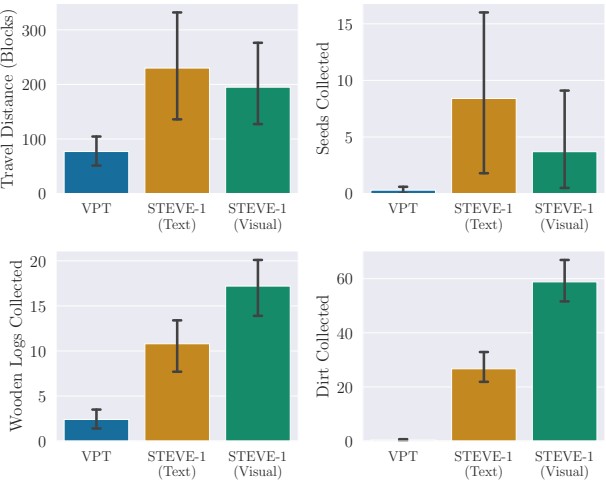
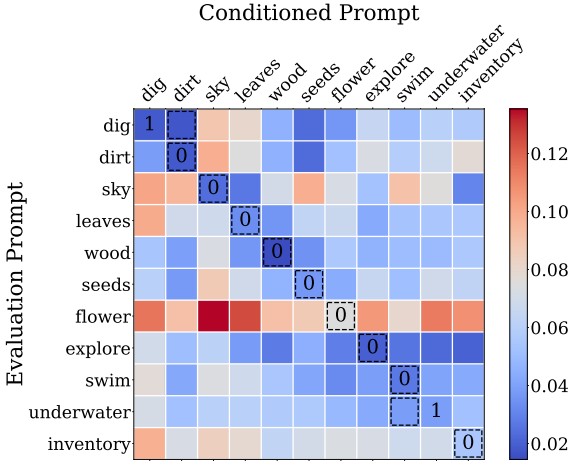

*Figure 3.* **Left:** In our programmatic evaluations, STEVE-1 performed far better than the unconditional VPT agent `early-game-2x` when prompted appropriately. On some tasks, visual outperforms text-based prompting, creating a gap that can likely be bridged through better prompt engineering. **Right:** Across our 11 MineCLIP evaluation tasks, STEVE-1 achieves the shortest distance between the episode and the MineCLIP goal embedding when prompted appropriately except for in two cases, where it mixes up digging and dirt and swimming and going underwater. This shows the strong general performance of STEVE-1 across a wide variety of short-horizon tasks. See Figure 14 for sample frames from each of the 11 visual goals.

the relevant text instruction, STEVE-1 collects 75x more dirt, 4.9x more wood, 22x more seeds, and travels 4.3x further than the unconditional agent. This also represents a significant improvement over the reported performance of text-conditioning in Appendix I of Baker et al. (2022), which collects several times fewer resources despite having twice as long of an episode to do so. We also run an automatic evaluation using MineCLIP embedding distances by measuring the minimum distance of a goal embedding to any frame in the episode. As shown in Figure 12, the distance between the goal and the episode is significantly lower when the agent is conditioned on the corresponding visual goal than otherwise. Full results for STEVE-1 with both text and visual goals can be found in Appendix F.

In addition to our evaluations of STEVE-1, we also recorded several sample interactive sessions we had with the agent (controlling it in real-time by giving it written text instructions or specific visual goals). These sessions demonstrate STEVE-1's ability to responsively follow instructions in real-time in a variety of situations. We believe that such use-cases, where humans give an agent natural instructions that it can follow to complete tasks, will become increasingly important and have practical uses in the creation of instructable assistants and virtual-world characters. These videos, as well as videos of our agent performing our evaluation tasks, can be found at https://sites.google.com/view/steve-1.

### 4.3. Prompt Chaining

We also experiment with longer horizon tasks that require multiple steps, such as crafting and building. We explore two different prompting methods: directly prompting with the target goal, and a simple form of prompt chaining (Chase, 2022; Wei et al., 2022b; Dohan et al., 2022) where the task is decomposed into several subtasks and the prompts are given sequentially for a fixed number of steps. We explore prompt chaining with visual goals for two tasks: 1) building a tower and 2) making wooden planks. When using prompt chaining, we first prompt STEVE-1 to gather dirt before building a tower, and to gather wooden logs before crafting wooden planks. Figure 4 shows that directly prompting STEVE-1 with the final tasks results in near-zero success rates. However, prompt chaining allows STEVE-1 to build a tower 50% of the time and craft wooden planks 70% of the time. For the tower building task, STEVE-1 immediately starts collecting dirt until the prompt switches, at which point its average height starts increasing rapidly and its dirt decreases as it builds a tower. Similarly, for the crafting wooden planks task, STEVE-1 immediately starts collecting a large amount of wooden logs until the prompt switches and it rapidly converts these wooden logs into wooden planks (causing the amount of wooden logs in its inventory to immediately decrease and the number of wooden planks to increase as it crafts more). Figure 4 visualizes the average item counts and agent height for the prompt chaining episodes. See Figure 17 and Figure 18 in the appendix for visualizations of specific prompt chaining

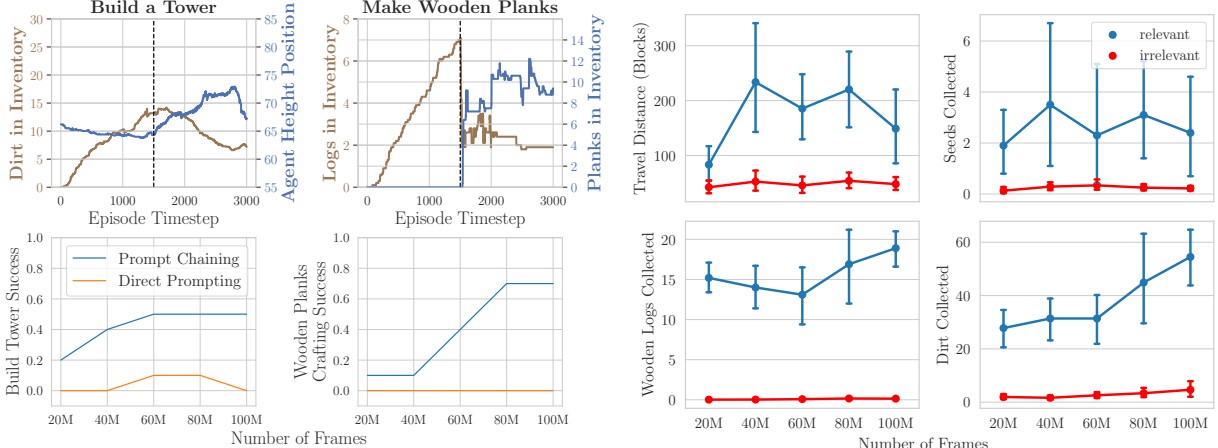

*Figure 4.* **Top left**: By sequentially chaining visual prompts like "get dirt" and "build a tower", STEVE-1 successfully gathers dirt and then uses this dirt to build a tower. The prompts switch at the dotted vertical line. **Bottom left**: The success rates of the chained prompts improve steadily as we train STEVE-1 on more data. **Right**: The performance of STEVE-1 in different tasks scales in different ways when conditioning on relevant visual prompt for the metric (e.g. break wood for the "Wooden Logs Collected" metric) versus other irrelevant visual prompts. For instance, in the wood-collection and dirt-collection tasks, performance starts increasing after training on 60M frames of gameplay.

episodes.

## 4.4. Scaling

Recent works in language modeling have found that scaling up pretraining FLOPs, by training on more data or by training a model with more parameters, can improve performance on downstream tasks (Kaplan et al., 2020; Suzgun et al., 2022; Wei et al., 2022a). In certain cases when measuring performance with metrics such as exact-match (Schaeffer et al., 2023), performance improvement may appear to be "emergent" (Wei et al., 2022a), appearing suddenly as the model is trained with more compute. Here, we aim to gain a basic understanding of how the performance of STEVE-1 on various tasks scales by training with more data (learning rate schedule is chosen appropriately).

To assess performance gain, we first isolated the performance of the policy from the prior, measuring performance of the agent through training on programmatic tasks (travel distance, seeds, logs, dirt) with visual goals. Due to compute constraints, we chose to use the 2x VPT model, which has 248M parameters. We found that both seed collection and travel distance did not improve significantly past 20M frames. From inspecting gameplay, we suspect that travel distance is a relatively easy task since it is close to VPT's default behavior of running around and exploring. For seed collection, performance remains suboptimal, suggesting that further scaling may be beneficial. This hypothesis is supported by the observation that performance on log and dirt collection remained roughly level until 60M frames when it

began to rapidly improve. Figure 4 shows the scaling curves for STEVE-1 on each programmatic task when conditioning on relevant vs. irrelevant visual prompts for that task.

We also evaluated the scaling properties of STEVE-1 for our multi-step tasks with and without prompt chaining. Without prompt chaining, the tasks remain challenging for STEVE-1 throughout training. However, we note that after 60M frames, STEVE-1 learns to gather wooden logs and build a small tower when told to build a tower. This is likely because our visual prompt for tower building shows a video of a tower being built out of wooden logs. With prompt chaining, the performance of STEVE-1 steadily increases with more data. We conjecture that this is because the success of a chained prompt requires the success of each element in the chain. Since different abilities emerge at different scales, one would expect chained prompts to steadily get more reliable as these subgoals become more reliably completed. In the case of building wooden planks, we note that crafting is one such task that gets significantly more reliable as the agent is trained on more data. Figure 4 shows the scaling curves for STEVE-1 on the prompt chaining tasks.

In summary, we see evidence of tasks that do not require much data for STEVE-1 to learn, tasks that steadily get more reliable as the agent is trained longer, and tasks where capability suddenly spikes after the agent reaches some threshold. Put together, this suggests that further scaling would likely significantly improve the agent, although we leave the task of predicting exactly how much performance there is to gain to future studies.

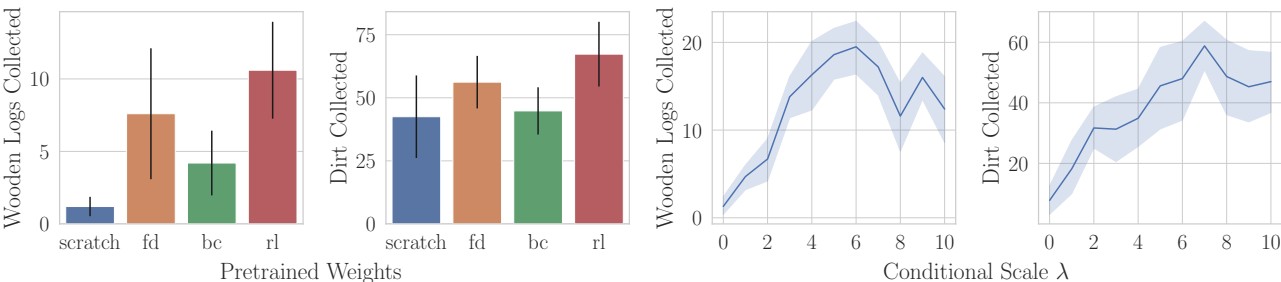

*Figure 5.* **Left:** We trained STEVE-1 on 100M frames starting from four different pretrained weights: random initialization (scratch), `foundation-2x` (fd), `bc-early-game-2x` (bc), and `rl-from-foundation-2x` (rl). The `rl-from-foundation-2x` agent is generally the most performant after fine-tuning. Using pretrained weights performs better than training from scratch, especially for more complicated tasks like collecting wood. **Right:** By using classifier-free guidance (Ho and Salimans, 2022), STEVE-1 collects 5.8x more dirt and 9x more wood.

### 4.5. What Matters for Downstream Performance?

In this section, we show that pretraining and classifier-free guidance both have a siginificant impact on downstream performance. See Appendix B for additional ablations on design choices for our method, including prompt engineering, the use of classifier-free guidance during training, text augmentation strategies, different VAE variants, and varying chunk sizes during finetuning.

**Pretraining** Baker et al. (2022) finds that by pretraining a behavioral prior with imitation learning on internet-scale datasets for Minecraft, the learned policy can be effectively finetuned to accomplish tasks that are impossible without pretraining. In this section, we demonstrate that pretraining is also massively beneficial for instruction-tuning in Minecraft. We hypothesize that due to the strong performance of STEVE-1 and the relatively small amount of compute ($\approx 1\%$ additional compute) used for instruction finetuning, most of the capabilities of our agent come from the pretraining rather than the finetuning. To test this hypothesis, we finetune several varients of STEVE-1 from various pretrained weights: `foundation-2x`, `bc-early-game-2x`, `rl-from-foundation-2x`, and with randomly initialized weights. In this experiment, each model was finetuned on 100M frames.

Figure 5 shows the performance of these models on our programmatic tasks with visual goals. Note that while an agent trained on our dataset from scratch can accomplish basic tasks like dirt collection fairly well, it is unable to find and chop down trees, in contrast to the pretrained agents. This demonstrates that the abilities present in the agent due to pretraining are successfully transferred to the fine-tuned agent. Out of all the pretrained weights we tried, we noticed that `rl-from-foundation-2x` performed the best, having qualitatively better performance at tasks like crafting and chopping down trees. Indeed, Figure 5 shows that this model has strong performance, likely due to the

massive amount of compute it was trained with during its RL training (Baker et al., 2022).

**Classifier-Free Guidance** Baker et al. (2022) observed that when conditioning the agent on text, it tended to ignore its instruction and instead perform the prior behavior learned during pretraining. As discussed in section 3.4, classifier-free guidance gives a knob for trading off between goal-conditioned and prior behaviors. Figure 5 shows the effect of this parameter $\lambda$ on the log and dirt collection tasks. The performance of the agent reaches its maximum around $\lambda = 5.0$ to $\lambda = 7.0$, after which it starts to drop off. These results demonstrate the importance of classifier-free guidance, which improves the performance of STEVE-1 by orders of magnitude.

## 5. Conclusion

We demonstrate the promise of our approach of applying unCLIP (Ramesh et al., 2022) and classifier-free guidance (Ho and Salimans, 2022) to create STEVE-1, a powerful instructable agent that is able to achieve a wide range of short-horizon tasks in Minecraft. STEVE-1 can be prompted to follow text or visual goals, achieving strong instruction-following performance in Minecraft with only $60 of compute and around 2,000 instruction-labeled trajectory segments. We show that an initial version of prompt chaining is a promising approach for improving performance on longer horizon tasks that require multiple steps of reasoning, and more can be done in future work to improve performance.

Due to the generality of our approach, which operates on raw pixels and produces low-level actions (mouse and keyboard), we hope that STEVE-1 can spark future work in creating instructable agents in other domains and environments. Future work should include addressing the limitations of STEVE-1 by improving its performance in longer-horizon tasks, perhaps through the use of LLMs.

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

## A. Broader Impact

With the increasing capability level of artificial intelligence comes many potential benefits and also risks. On the positive side, we anticipate that the techniques that used to create STEVE-1 could be applied to the creation of helpful agents in other sequential decision making domains, including robotics, video games, and the web. Our demonstration of such a low cost approach to creating a powerful, instruction-following model also has the potential to improve the democratization of AI. However, on the negative side, agents pretrained on large internet datasets reflect the biases of the internet and, as suggested by our experiments, these pretraining biases can potentially remain after instruction-tuning. If not addressed carefully, this could lead to devastating consequences for society. We hope that while the stakes are low, works such as ours can improve access to safety research on instruction-following models in sequential decision-making domains.

## B. Additional Ablations

In this section, we describe additional ablations on design choices for our method, including prompt engineering, the use of classifier-free guidance during training, text augmentation strategies, different VAE variants, and varying chunk sizes during finetuning. We use programmatic evaluation metrics to compare the performance of the various ablations.

### B.1. Prompt Engineering

Prompt engineering as a discipline has rapidly emerged over the last year due to the observation that the quality of the output of text-to-X models can dramatically change depending on the prompt (Zhou et al., 2023). For example, Table 1 shows how a prompt for Stable Diffusion (Rombach et al., 2022) might be written. By listing out the various attributes of the image such as visual medium, style, and the phrase "trending on ArtStation", the user is able to get a higher quality image (Gustavosta, 2023; Liu and Chilton, 2022). In this section, we explore how this same style of prompt engineering can improve the performance of STEVE-1. Figure 6 shows how a simple prompt of "get dirt" might be changed in order to more accurately specify the type of behavior that is desired. Just like in image generation models, the performance of STEVE-1 significantly improves by modifying the prompt in this fashion. By changing to more complicated prompts, STEVE-1 is able to collect 1.56x more wood, 2x more dirt, and 3.3x more seeds.

| Prompt | Dirt Collected |
|---|---|
| "break a flower" | 0.7 (-0.2, 1.6) |
| "collect seeds" | 2.7 (0.9, 4.5) |
| "dig as far as possible" | 3.9 (2.8, 5.0) |
| "get dirt" | 9.2 (5.7, 12.7) |
| "get dirt, dig hole, dig dirt, gather a ton of dirt, collect dirt" | **26.7 (19.9, 33.5)** |

*Figure 6.* Similar to in image generation, switching to a longer, more specific prompt dramatically improves the performance of STEVE-1.

| Model | Simple Prompt | Complex Prompt |
|---|---|---|
| Stable Diffusion (Rombach et al., 2022) | steampunk market interior | steampunk market interior, colorful, 3D scene, Greg Rutkowski, Zabrocki, Karlkka, Jayison Devadas, trending on ArtStation, 8K, ultra-wide-angle, zenith view, pincushion lens effect (Gustavosta, 2023) |
| STEVE-1 | collect seeds | break tall grass, break grass, collect seeds, punch the ground, run around in circles getting seeds from bushes |

*Table 1.* Example of evolving simple prompts into more complex ones for various models.

## B.2. Classifier-Free Guidance During Training

We examine the importance of using classifier-free guidance during training by finetuning a model with *no* guidance which does not drop out the goal embedding $z_{\tau_{goal}}$ from the policy's input (i.e., $p_{uncond} = 0.0$) and comparing it to the version which uses guidance ($p_{uncond} = 0.1$). The chunk size is set to the range 15 to 50 and we train each policy for 100M frames. In Figure 7, we compare the performance of using *visual* goals (MineCLIP video embedding) on the no guidance model using conditional scale $\lambda = 0$ and the guidance model using conditional scales $\lambda = 0$ and $\lambda = 3$. We observe that while the no guidance model slightly outperforms the guidance model at $\lambda = 0$ across a few metrics, the agent with guidance outperforms the no guidance agent by a factor of 2 to 3 times for the inventory collection tasks when we increase the conditional scale to $\lambda = 3$ (which we cannot do for the no guidance model). For the travel distance metric, both of the guidance versions perform similarly to the no guidance version.

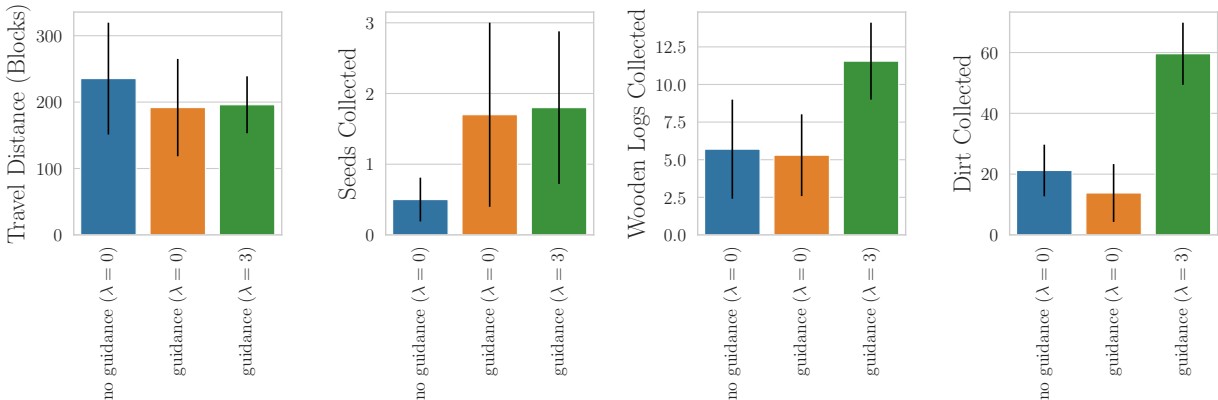

Figure 7. **Ablation on Guidance**. In the "*no guidance*" variant, we set $p_{uncond} = 0$, meaning that we do not drop any $z_{\tau_{goal}}$ from the policy's input during training. The "*guidance*" variants set $p_{uncond} = 0.1$, dropping 10% of the time during training. Whereas the "*no guidance*" model is only compatible with $\lambda = 0$ at inference, the "*guidance*" model can use $\lambda > 0$, allowing for better performance.

## B.3. Text Augmentation

During finetuning, instead of using only self-supervision with future MineCLIP video embedding as the goal, we considered using the *text* embeddings from the 2,000 human labeled trajectory segments as goal embeddings, either solely or in addition to the self-supervised video embeddings. In order to more fairly compare with the CVAE prior approach, we augment the human-labeled data with additional text-gameplay pairs generated as described in Appendix E.2. We implement this experiment by replacing the visual embeddings used for relabeling in Algorithm 1 with text embeddings, when available, with a 90% probability. To experiment with not using visual embeddings at all, we can replace the visual embeddings with zeros in the same way. In Figure 8, we observe that using only the visual embeddings during training, in combination with the CVAE, can outperform using MineCLIP text embeddings directly in the other two baselines. In this experiment, the chunk size is set to the range 15 to 50 and we train each policy for 100M frames.

## B.4. VAE Variants

We study the dataset used to train the CVAE prior model. In Figure 9, we observe that augmentation helps in some programmatic tasks, including the dirt and seed collection tasks, but slightly hurts the wooden log collection and travel distance metrics. In this experiment, we use the same policy with each CVAE variant and we tune the conditional scale $\lambda$ for each variant. The chunk size is set to the range 15 to 200 and we train the policy for 100M frames.

## B.5. Chunk Size

During finetuning, we compare different goal chunk sizes by varying the `max_btwn_goals=[100,200,300,400]`, while keeping the `min_btwn_goals=15`. See Algorithm 1 for more details. A larger `max_btwn_goals` introduces more noise, with actions that led to achieving the further away goal being less correlated to the actions present in that goal chunk. In Figure 10, we observe that the best `max_btwn_goals` chunk size is around 200, and increasing the chunk size beyond that causes a drop in performance. We train each policy for 160M frames and tune the conditional scale for each.

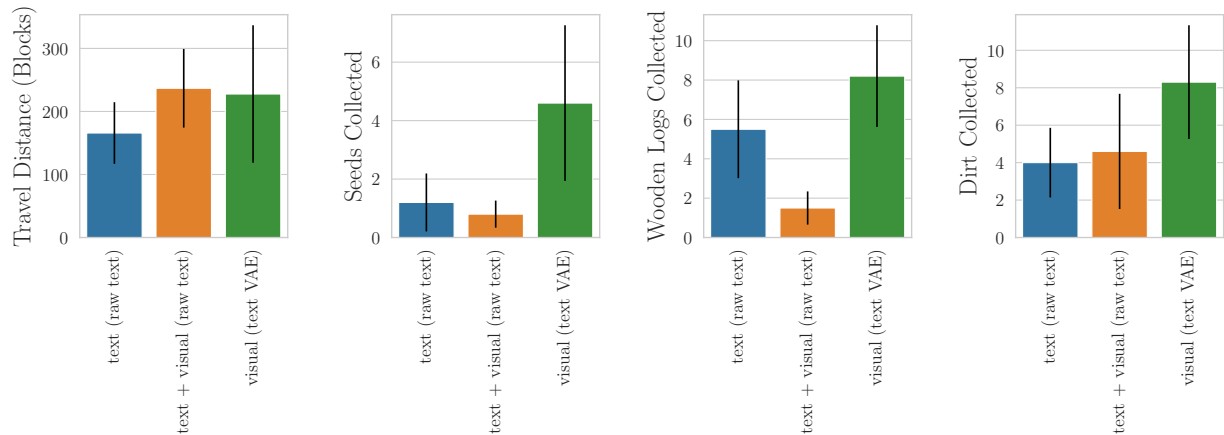

*Figure 8.* **Ablation on Text Augmentation**. In the "*text (raw text)*" ablation, we train the model using only the text labels from human labelled trajectory segments, and directly use the MineCLIP text embedding of the text label as the goal embedding during training and at inference. For the "*text + visual (raw text)*" ablation, we use both the visual embedding in self-supervised manner and the text embedding from the human labelled trajectory segments during training and use the MineCLIP text embedding during inference. Even with augmentation, the dataset only contained around 2% text embeddings. The "*visual (text VAE)*" version is as reported in the main method, using the CVAE to convert MineCLIP text embedding to visual embedding during inference.

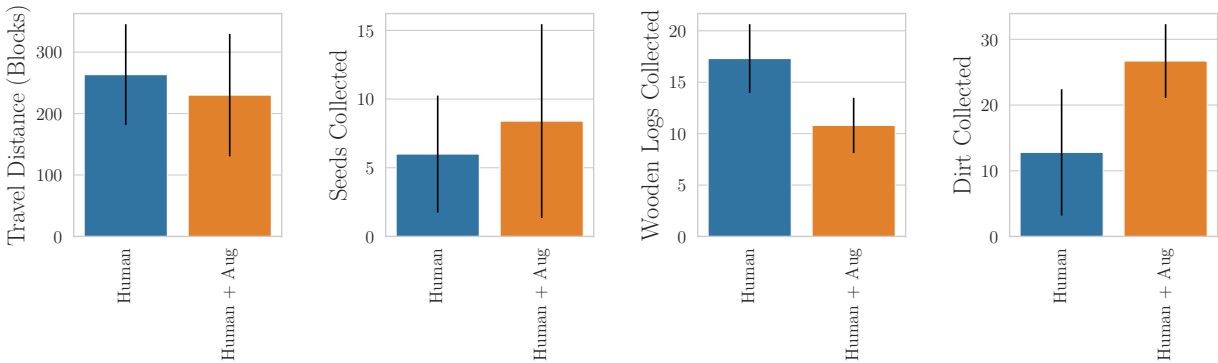

*Figure 9.* **Ablation on VAE Training Data**. "*Human*" baseline uses only the 2,000 human-labelled trajectory segments (text-video pairs), as training example for the CVAE prior model. "*Human + Aug*" baseline adds additional pairs of text-video examples as described in Section 3.3.

## C. Dataset Details

### C.1. Gameplay Dataset

Our gameplay dataset consists of two types of episodes: 7,854 episodes (38.94M frames) of a contractor dataset made available from Baker et al. (2022) and 2,267 episodes (14.96M frames) of gameplay generated by running various pretrained VPT agents.

**OpenAI Contractor Dataset**    The majority of our data comes from the contractor data used to train VPT (Baker et al., 2022). OpenAI released five subsets of contractor data: 6.x, 7.x, 8.x, 9.x, and 10.x. We use an equal mix of 8.x, 9.x, and 10.x, which correspond to "house building from scratch", "house building from random starting materials", and "obtain diamond pickaxe". Contractors were given anywhere from 10 to 20 minutes to accomplish these goals to the best of their abilities while their screen, mouse, and keyboard were recorded.

**VPT-Generated Dataset**    We generated additional data by generating episodes using various pretrained VPT agents. In order to increase the diversity of data as well as to get data of the agent switching tasks randomly throughout the middle of episodes, we added random switching between the different pretrained agents during episodes. Specifically, at the

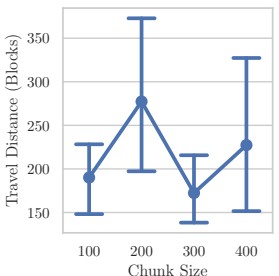 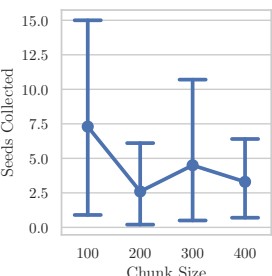 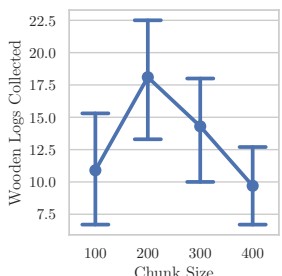 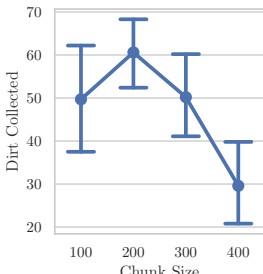

*Figure 10.* **Ablation on Segment Chunk Size**. We vary the `max_btwn_goals` parameter in Algorithm 1. The performance is roughly the best at around 200, beginning to decline with greater values.

beginning of an episode we randomly sample two VPT agents from (`foundation_model_2x`, `bc_early_game_2x`, `bc_house_3x`, `rl_from_foundation_2x`, `rl_from_house_2x`) and switch between them at each timestep with a probability of $1/1000$. Since the RL agents all act quite similarly, we avoid sampling two RL agents at once. Additionally, with a probability of $1/750$ each timestep, we cause the agent to spin a random number of degrees. This adds more data where the agent spontaneously changes tasks, increasing downstream steerability.

### C.2. Instruction Dataset

We gathered a small dataset of 2,000 human-labelled trajectory segments (text-video pairs) by manually labeling gameplay from our datasets. We used a simple web app that presented a video of 16 frames to the user from a randomly sampled episode. This only corresponds to 32,000 frames of labeled data, which corresponds to labeling 0.06% of the full dataset, or 27 minutes of labeled data. However, as discussed in section E.2, combining this with automatically labeled data using `gpt-3.5-turbo` and MineCLIP results in a strong prior model.

---

**Algorithm 1** Sampling Episode Segments with **Packed Hindsight Relabeling**

---

**Function** `sample_episode_segment` (*T, min_btwn_goals, max_btwn_goals*)

    segment = sampleSegment(episode, T)

    curr_timestep = segment.start

    goal_switching_indices = []

    **while** curr_timestep < segment.end **do**

        curr_timestep += uniform(min_btwn_goals, max_btwn_goals)

        goal_switching_indices.append(curr_timestep)

    relabeled_goal_embeds = []

    **for** n in range(1, len(goal_switching_indices)) **do**

        relabeled_goal_embeds[$i_{n-1}$:$i_n$] = segment.goal_embeddings[$i_n$]

    **return** segment.obs, segment.actions, relabeled_goal_embeds

---

## D. Evaluation Details

Evaluating the performance of our agent is a challenging task due to the wide variety of instructions that are possible and the difficulty of evaluating whether the agent has successfully achieved its task. We use a combination of programmatic evaluation metrics and automatic MineCLIP evaluation metrics to get a sense of the agent's capability level.

**Programmatic Evaluation** We compute programmatic evaluation metrics by monitoring the MineRL (Guss et al., 2019) environment state throughout each evaluation episode. As done in VPT (Baker et al., 2022), we compute multiple programmatic metrics including travel distance and early-game item collection. The travel distance is the maximum displacement of the agent along on the horizontal (X-Z) plane, measured from the initial spawn point. For early-game inventory counts, we store the maximum number of log, seed, and dirt items seen in the agent's inventory during the episode.

**MineCLIP Evaluation**    For any trajectory $\tau$ we can use the text-visual alignment of MineCLIP embeddings (Fan et al., 2022) to roughly evaluate whether a segment of 16 frames corresponds to a given task. We explore the use of alignment in the MineCLIP latent space between trajectories and either text or visual goals to evaluate our agent over a wider variety of tasks where programmatic evaluation isn't practical. To determine the degree to which a task has been completed at all during an episode, we record the minimum cosine distance between the (text or visual) goal embeddings at any timestep during an episode.

## E. Training Details

### E.1. Policy Training

STEVE-1 was trained using distributed data parallel in PyTorch (Paszke et al., 2019). During training, segments of 640 timesteps were sampled from the dataset. Due to memory constraints, these segments were further broken up into chunks of 64, which are processed sequentially. Since VPT uses a Transformer-XL (Dai et al., 2019), this sequential processing lets the policy attend to previous batches up to the limit of its context length. We optimized the weights using AdamW (Loshchilov and Hutter, 2019) with a maximum learning rate of 4e-5 and a linear warmup for the first 10M frames followed by a cosine learning rate decay schedule that decays to 10% of the original learning rate. See Table 3 for an exhaustive list of hyperparameters used during training.

During training, we sample data using packed hindsight relabeling (Figure 2). This involves sampling a segment of an episode, randomly selecting some timesteps at which to change goals, and then filling in the corresponding goal embeddings for the entire episode with the embeddings from the corresponding goal segments. See Algorithm 1 for a detailed explanantion of packed hindsight relabelling.

| Hyperparameter Name | Value |
|---|---|
| `architecture` | MLP |
| `hidden_dim` | 512 |
| `latent_dim` | 512 |
| `hidden_layers` | 2 |
| `batch_size` | 256 |
| `learning_rate` | 1e-4 |
| $\beta$ | 0.001 |
| `n_epochs` | 50 |
| `n_search_episodes` | 2000 |
| `k` | 5 |
| `offset` | 8 |

*Table 2.* Prior Hyperparameters

### E.2. Prior Training

The prior model is a simple CVAE (Sohn et al., 2015) that conditions on MineCLIP (Fan et al., 2022) text embeddings and models the conditional distribution of visual embeddings given the corresponding text embedding. This model is trained on a combination of around 2,000 hand-labeled trajectory segments and augmented with additional data by automatically searching for text-gameplay pairs from our gameplay dataset. This is done using the following steps:

1. Combine the 2,000 text labels with 8,000 additional labels generated by querying `gpt-3.5-turbo`.

2. For each of these 10,000 text labels, search through 1,000 episodes sampled from the gameplay dataset to find the top 5

| Hyperparameter Name | Value |
|---|---|
| `trunc_t` | 64 |
| `T` | 640 |
| `batch_size` | 12 |
| `num_workers` | 4 |
| `weight_decay` | 0.039428 |
| `n_frames` | 160M |
| `learning_rate` | 4e-5 |
| `optimizer` | AdamW (Loshchilov and Hutter, 2019) |
| `warmup_frames` | 10M |
| `p_uncond` | 0.1 |
| `min_btwn_goals` | 15 |
| `max_btwn_goals` | 200 |
| `vpt_architecture` | 2x |

*Table 3.* Policy Hyperparameters

closest visual MineCLIP embeddings to the text embedding of the text label.

These 50,000 automatically-mined text-video pairs are added to the original 2,000 hand-labeled examples to form the final dataset used for prior training.

We noticed when prompting STEVE-1 using visual goals that when the visual goal showed the agent hitting a block but not following through and breaking it that STEVE-1 actually avoided breaking blocks. Unfortunately, many of the automatically discovered text-gameplay clips include gameplay of this kind. In order to prevent this issue, we added an offset to the embeddings found in this manner. By selecting embeddings from a timestep `offset` steps after the originally-selected timestep, the agent is much more likely to follow through with breaking blocks.

We trained our prior model for 50 epochs on this dataset and used early-stopping with a small validation set. An exhaustive list of hyperparameters used for creating the prior model can be found at Table 2.

## F. Additional Visualizations

### F.1. MineCLIP Evaluation

We ran MineCLIP evaluation on both text and visual prompts. The MineCLIP evaluation results can be found in Figure 13.

### F.2. Steerability with Programmatic Metrics

Similar to Figure 20 in the VPT appendix (Baker et al., 2022), we plot the programmatic metric performances (mean and 95% confidence intervals) across the different goal prompt conditioning, both using visual prompts (Figure 15) and text prompts with CVAE prior (Figure 16) conditioning, on our policy trained with hyperparameters in Table 3 and using conditional scaling $\lambda = 7$ (for visual prompts) and $\lambda = 6.0$ (for text prompts with CVAE prior). Each conditioning variant is run with 10 trials, each trial with a different environmental seed and with an episode length of 3000 timesteps (2.5 minutes gameplay). Across the conditioning variant, we use the same set of environmental seeds. For comparison, we also plot the metrics for an unconditional VPT (`early_game`) agent ("*VPT (uncond)*") and the text-conditioned agent investigated in VPT appendix (Baker et al., 2022) ("*VPT (text)\**") when conditioned on the relevant text. When using *visual*

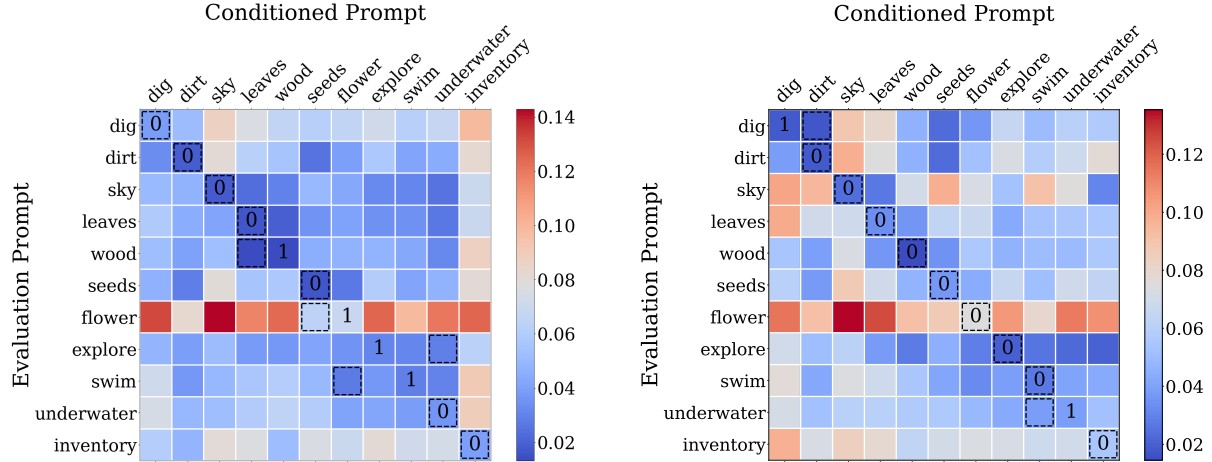

*Figure 11.* MineCLIP Text Evaluation          *Figure 12.* MineCLIP Visual Evaluation

*Figure 13.* **MineCLIP Evaluation**. We measure the cosine distance between the goal embedding given to the agent and the MineCLIP video embeddings throughout the episode and record the minimum across the episode. Dashed box indicates the minimum along the row, and the number in the diagonal box indicates the rank of the diagonal element (0 is minimum) in the row. **Left:** We use the prior to convert the text into the goal embedding. Across our 11 text MineCLIP evaluation tasks, STEVE-1 achieves the shortest distance between the episode and the MineCLIP goal embedding when prompted appropriately for most cases. This shows the strong general performance of STEVE-1 across a wide variety of short-horizon tasks. **Right:** We embed the visual goal loops (Figure 14) with MineCLIP video encoder. Across our 11 visual MineCLIP evaluation tasks, STEVE-1 achieves the shortest distance between the episode and the MineCLIP goal embedding when prompted appropriately except for in two cases, where it mixes up digging and dirt and swimming and going underwater. This shows the strong general performance of STEVE-1 across a wide variety of short-horizon tasks.

goal conditioning, we the use MineCLIP video encoder to embed a 16-frame clip of the agent performing the desired task taken from our training dataset. An example frame from each of the visual goals is illustrated in Figure 14. When using *text VAE* goal conditioning, we use the MineCLIP text encoder to encode the text prompts (Table 4) and use the CVAE prior to sample the goal embedding from the MineCLIP text embedding.

We note several differences in our experimental setup compared to that in VPT (Baker et al., 2022). We only run our evaluation episodes for 3000 timesteps, equivalent to 2.5 minutes of gameplay, compared to 5 minutes in the VPT paper. Due to a limited computational budget, we generate 10 episodes per conditioning variant, and 110 episodes for the unconditional ("*VPT (uncond)*"), compared to VPT's 1000 episodes. Lastly, when measuring the inventory count, we log the maximum inventory count seen throughout the episode, which is a lower bound on the potential number of items collected since the agent can later throw out, place, or use these items to craft. As a result of these caveats, we denote the "*VPT (text)\**" legend in Figure 15 and Figure 16 with an asterisk as we use the results reported in (Baker et al., 2022) directly for comparison.

We make several observations. First, we observe that our agents is more *steerable*: when conditioned to collect certain items (in bold), the agent collects (relatively) many more of those items than when conditioned on other instructions unrelated to that item, as well as compared to the unconditional VPT. When conditioned on tasks unrelated to the item (e.g. break a flower when interested in measuring logs collected), we also observe that the agent pursues that item *less* than the unconditional agent. Second, we observe that for the bolded instructions which we expect to stand out, we outperform VPT performance (dashed blue line) (Baker et al., 2022), even with half the amount of time in the episode rollout. This suggests that our agent is both more steerable relative to the unconditioned VPT agent and the text-conditioned VPT agent investigated in the VPT appendix (Baker et al., 2022).

### F.3. Prompt Chaining Visualization

We visualize two specific episodes from the prompt chaining experiments in Section 4.3 in Figure 17 and Figure 18.

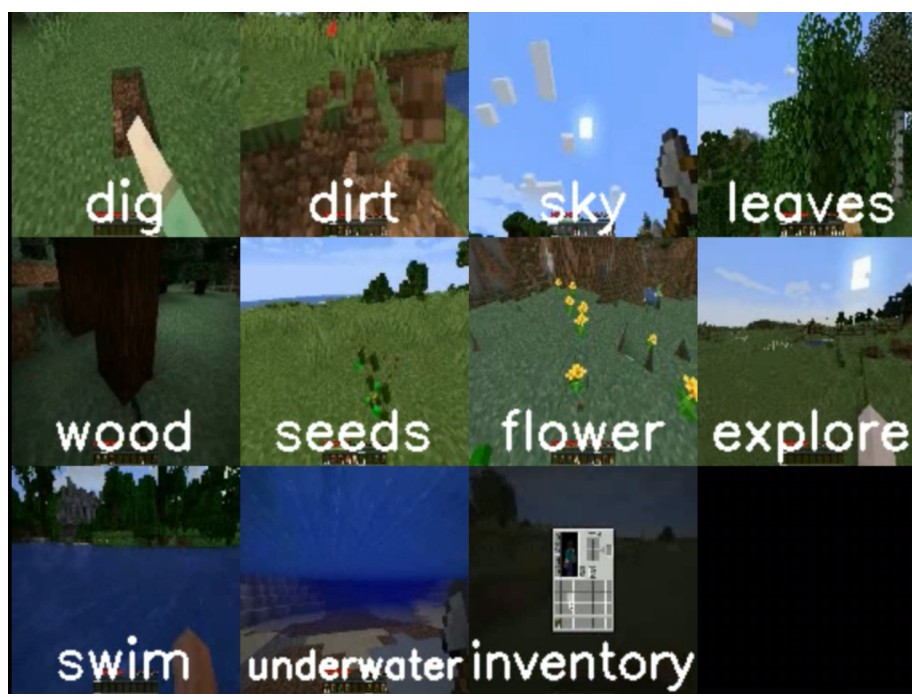

*Figure 14.* Sample frames from each of the 11 visual goals. Note that the text overlaid on the frame is *not* present when we encode the 16-frame clip with MineCLIP video encoder, and is only present for the figure visualization.

| Conditioning Variant Name | Text Prompt |
| --- | --- |
| dig as far as possible | dig as far as possible |
| get dirt | get dirt |
| look at the sky | look at the sky |
| break leaves | break leaves |
| chop a tree | chop a tree |
| collect seeds | collect seeds |
| break a flower | break a flower |
| go explore | go explore |
| go swimming | go swimming |
| go underwater | go underwater |
| open inventory | open inventory |
| get dirt . . . | get dirt, dig hole, dig dirt, gather a ton of dirt, collect dirt |
| chop down the tree . . . | chop down the tree, gather wood, pick up wood, chop it down, break tree |
| break tall grass . . . | break tall grass, break grass, collect seeds, punch the ground, run around in circles getting seeds from bushes |

*Table 4.* Strings corresponding to each conditioning variant for the text VAE conditioning. For conditioning variants without "...", the text prompt is the same as the conditioning variant name.

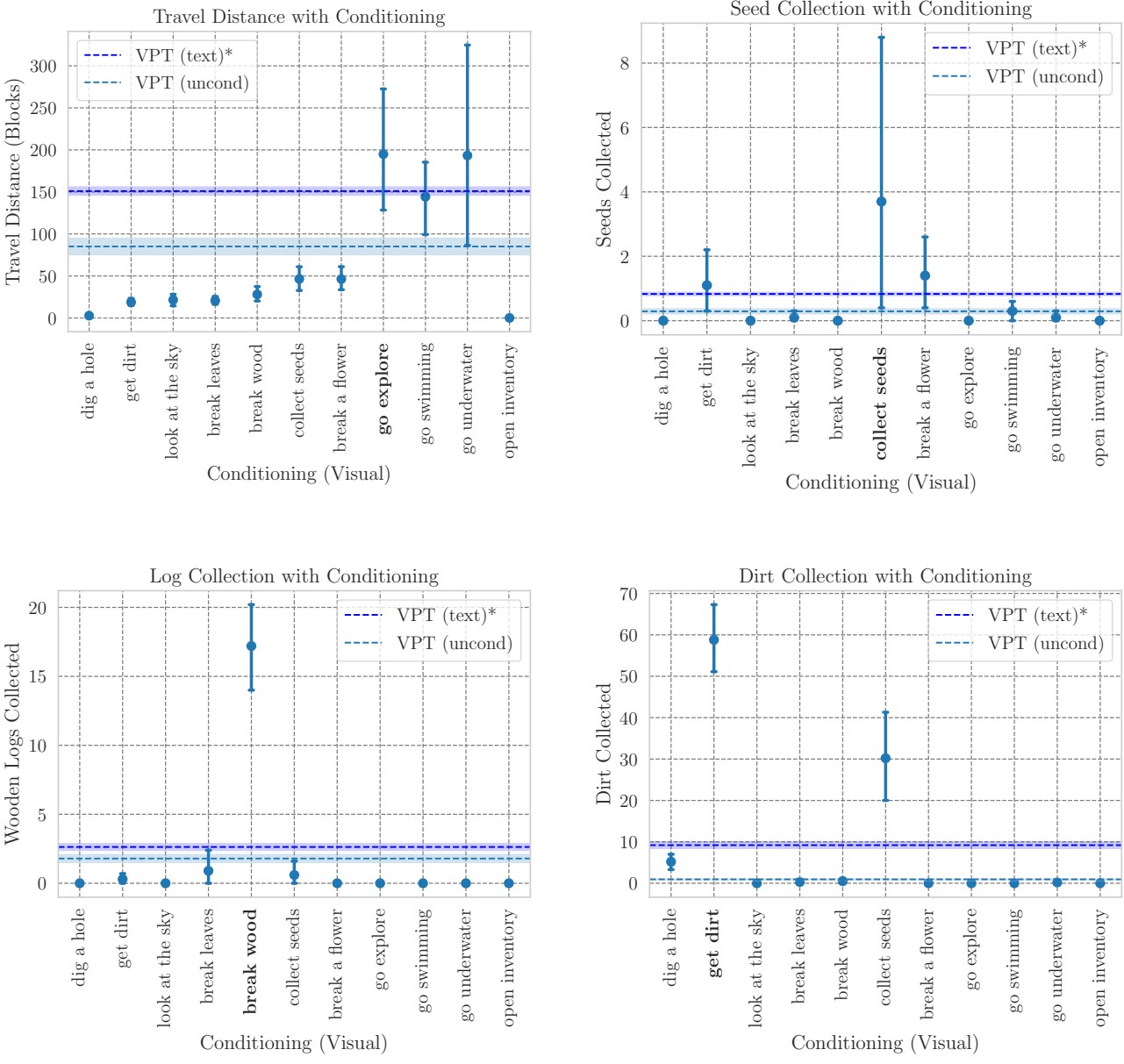

*Figure 15.* **Conditioning with Visual Goals**. We plot the performance of the programmatic metrics, along with their mean values and 95% confidence intervals, across different goal conditioning. See Figure 14 for visualization of these visual prompts. Plots are similar to Figure 20 in the VPT appendix (Baker et al., 2022). Each conditioning variant is run with 10 trials, each with a different environmental seed and with an episode length of 3000 time steps (2.5 minutes gameplay). We use the policy that was trained using the hyperparameters specified in Table 3, and with conditional scaling values $\lambda = 7$. The dashed horizontal lines refer to an unconditional VPT agent ("*VPT (uncond)*") and a text-conditioned agent from the VPT appendix ("*VPT (text)\**") that was conditioned on the relevant text, for the purpose of comparison.

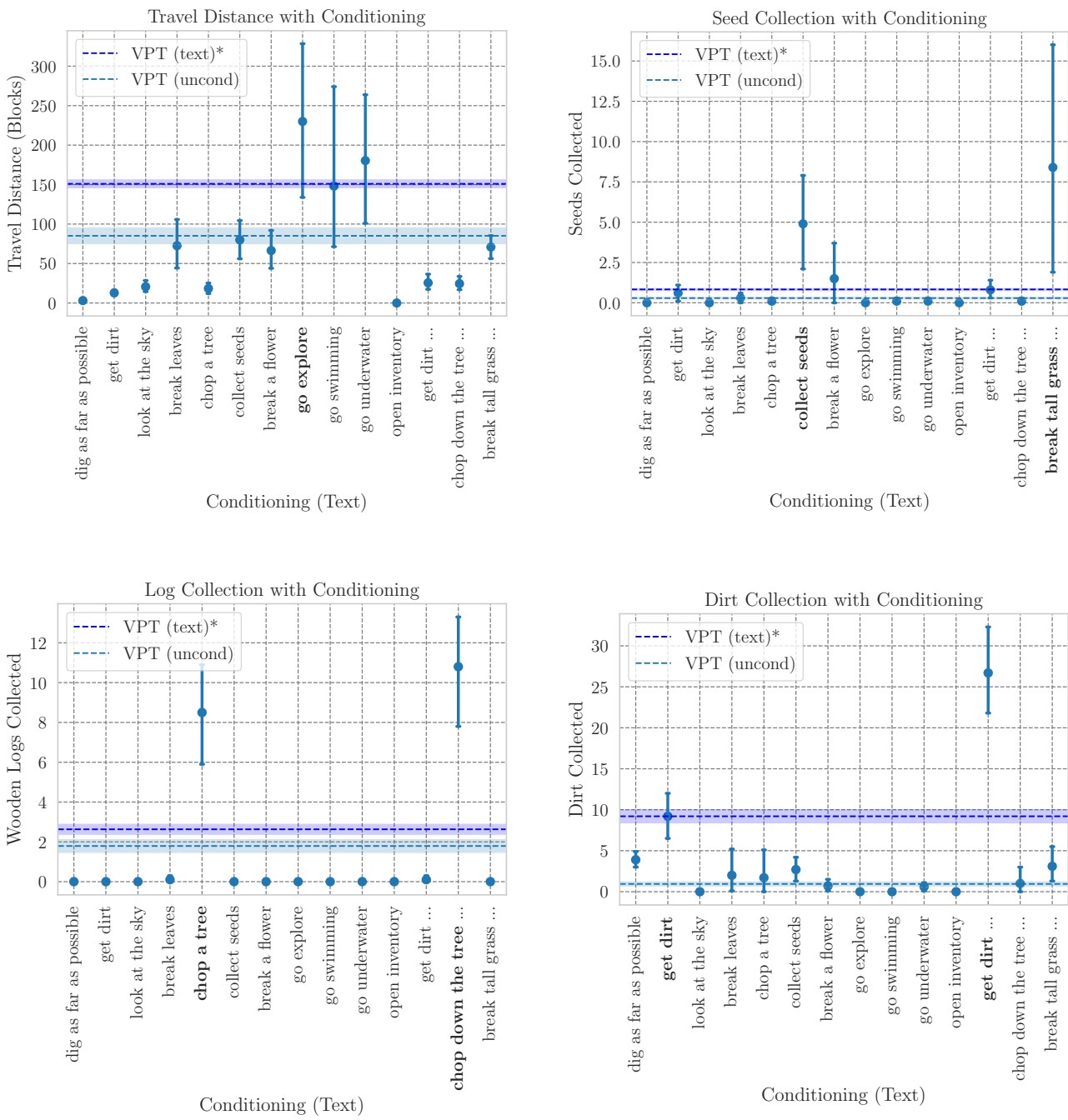

*Figure 16.* **Conditioning with text goals**. See Table 4 for the text string used for each conditioning variant. We use the same policy model but with a conditional scaling value $\lambda = 6$. We observe strong steerability which outperforms VPT text conditioning in (Baker et al., 2022), and we observe that more specific prompt engineered text can lead to improvements.

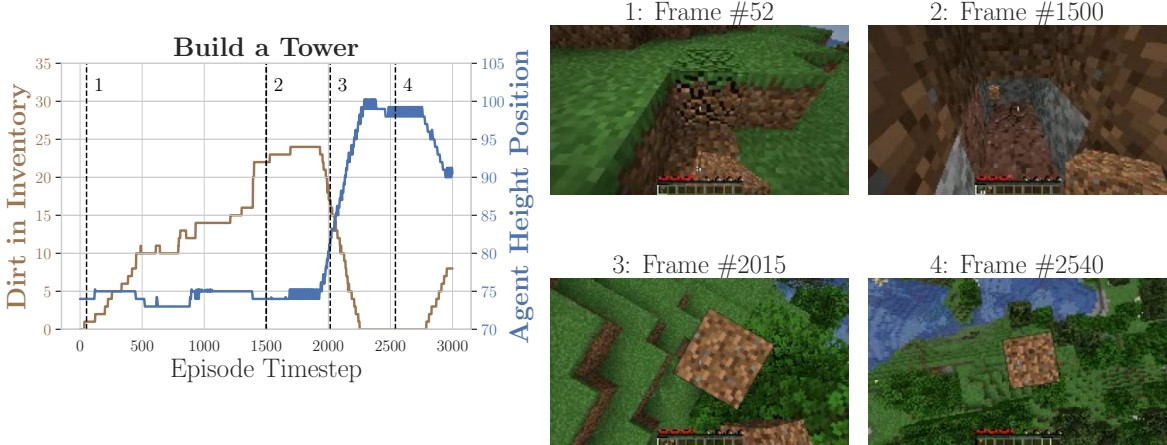

*Figure 17.* **Build a Tower task**. (*Left*) We track the amount of dirt in the inventory and the agent's height position (y-axis) throughout the episode. In the first 1500 timesteps, the agent is conditioned on the visual get dirt goal, then the agent is conditioned on the visual build a tower goal for the final 1500 timesteps. Vertical dotted lines with numbers indicate the corresponding frames on the right. (*Right*) The agent's observation frames at 4 different points in the episode. First the agent collects dirt, then begins to build the tower using the dirt blocks.

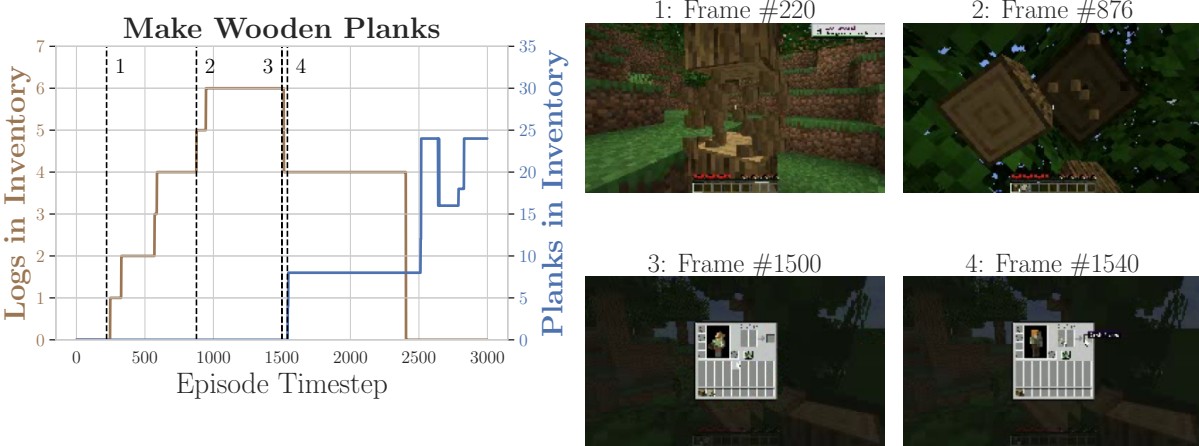

*Figure 18.* **Make Wooden Planks task**. (*Left*) We track the number of logs and planks in the inventory. In the first 1500 timesteps, the agent is conditioned on the visual break wood goal, then the agent is conditioned on crafting the visual wooden planks goal for the final 1500 timesteps. Similarly to Figure 17, a vertical dotted line annotated with a number indicates the corresponding frame to the right. (*Right*) The agent's observation frames at 4 different points in the episode. First the agent breaks trees to collect wooden logs, then opens the inventory and crafts wooden planks.

