# OpenReview forum: "A Generative Model for Text Control in Minecraft"
_ICML.cc/2023/Workshop/ILHF — ILHF Workshop ICML 2023_

### Official Review · Reviewer_FtvR · 2023-06-15
**Timely Work**

**Rating:** 8
**Confidence:** 4

**Review:**

This paper proposes an instruction-following generative model for the game Minecraft, called STEVE-1, which is trained to follow text instructions by first mapping them to a latent goal, and then auto regressively sampling actions for the game agent to follow.

Pros:
1. Very timely work, combining recent interest and success in instruction-tuning models and foundation models for sequential decision making problems
2. Minecraft is a popular test-bed and relevant domain
3. Achieves strong results and offers a strong set of ablations

Cons:
1. The paper currently makes it hard to see examples of text instructions. It would be nice if there was either a central figure or a table in the main paper with clear examples of the time of text instructions STEVE-1 works well on ( Fig 3 is hard to understand, is a prompt one word?)

---

### Decision · Program_Chairs · 2023-06-20

Accept